

# Prevalence of sarcopenia was higher in women than in men: a cross-sectional study from a rural area in eastern China

Yichen Yang, Qin Zhang, Caihong He, Jing Chen, Danfeng Deng, Wenwen Lu and Yuming Wang

Department of Geriatrics, The First Affiliated Hospital, College of Medicine, Zhejiang University, Hangzhou, Zhejiang, China
Key Laboratory of Diagnosis and Treatment of Aging and Physic-chemical Injury Diseases of Zhejiang Province, The First Affiliated Hospital, College of Medicine, Zhejiang University, Hangzhou, Zhejiang, China

## ABSTRACT

**Background**. There were limited studies specifically evaluating whether the difference of the prevalence of sarcopenia exists in men and women in older adults from rural areas in China. The aim of this study was to compare the prevalence of sarcopenia between men and women in a rural area in eastern China and to explore the underlying causes.
**Methods**. This study included 1,105 participants aged 60-89 years. Muscle mass was measured by bio-electrical impedance analysis. Hand grip strength was measured by Jamar Hydraulic Hand Dynamometer. Sarcopenia was diagnosed according to the Asian Working Group for Sarcopenia-2019 Consensus. Data were analyzed using log-binomial and linear regression.
**Results**. The prevalence of sarcopenia was 21.7% in women and 12.9% in men among the study cohort. After adjusting for age, education level, number of diseases, income level, smoking, drinking, and eating habits, proportion of people with sarcopenia was 1.49-fold greater in women than in men (PR = 1.49, 95% CI [1.01–2.26], $P = 0.055$).
**Conclusions**. The prevalence of sarcopenia in elderly women in this rural area of eastern China is higher than in men, suggesting that women in rural areas in China seem to be more vulnerable for sarcopenia, thus early screening and prevention need to be provided for them to address such gender disparity in health.

## INTRODUCTION

Sarcopenia is an aging-related syndrome characterized by progressive loss of skeletal muscle mass and strength, which would lead to disability, decline in quality of life and even death in older adults (*Dennison, Sayer & Cooper, 2017*), thus increasing the burden of family and social medical care (*Beaudart et al., 2017*). Despite difference in populations, worldwide, the overall prevalence of sarcopenia in the elderly people over 60 years is about 10% (*Shafiee et al., 2017*). Globally, the prevalence of sarcopenia was higher in men compared with women when the European Working Group on Sarcopenia in Older People 2 (EWGSOP2) (11.0% vs. 2%) and muscle mass (35% vs. 27%) were used for classification. Women classified using the International Working Group on Sarcopenia (IWGS) had a higher prevalence

Corresponding author
Qin Zhang, zhangqin1978@zju.edu.cn

of sarcopenia than men (17% vs. 12%) while the prevalence by sex was similar using the EWGSOP, the Asian Working Group for Sarcopenia (AWGS), and the Foundation for the National Institutes of Health (FNIH) (*Petermann-Rocha et al., 2022*). To name a few, among the community elderly in Thailand, the prevalence of sarcopenia was higher in women (*Yuenyongchaiwat & Akekawatchai, 2022*); sex difference in the prevalence of sarcopenia varied among Brazilian community residents using different diagnostic criteria and cut-off values (*Fernandes et al., 2021*). In terms of sarcopenia among urban residents in China, several studies have been completed, with some finding a higher prevalence of sarcopenia among urban elderly men than among women (*Chen et al., 2021*; *Hai et al., 2017*; *Liu et al., 2020*), while others finding the opposite results (*Gao et al., 2015*; *Wang et al., 2016*; *Wang et al., 2019*). The population aging process is serious in rural areas of China (*Yu, Liu & Wang, 2018*), where the population has lower education level, less access to quality healthcare (*Dong et al., 2020*), lower incomes and poor health condition (*Dai, Zhou & Mei, 2013*) compared to those living in urban areas. To date few studies have specifically evaluated whether the difference of the prevalence of sarcopenia exists in men and women in older adults from rural areas in China, especially rural areas in eastern China. In a previous study of sarcopenia in urban and rural areas of Western China, the dietary status of the population and its effect on sarcopenia were not investigated and analyzed (*Gao et al., 2015*), which is exactly what we want to consider. From another perspective, although women have longer life expectancy than men (*Fang et al., 2015*), they have higher disease incidence rate and disability level, and studies have shown that the health status of elderly women is worse than that of old men (*Burton-Jeangros & Zimmermann-Sloutskis, 2016*; *Liu et al., 2016*). Based on the higher prevalence of sarcopenia in women than in men noted in some studies, we conducted a prevalence survey of sarcopenia in a rural area in eastern China to identify whether women are vulnerable population for sarcopenia so that tailored intervention could be provided.

The aim of this study was to compare the prevalence of sarcopenia between men and women in rural areas of Doumen Town, Shaoxing City, Zhejiang Province, China, and to explore the related factors for this difference.

## MATERIALS & METHODS

### Study participants

The data are from a national key research and development program (No. 2018YFC2000301), which is a national multi-center study assessing the characteristics of health status changes in the process of aging in Chinese population. The participants were recruited from the local community and clinic centers by distributing posters. From May to August 2019, we recruited people aged 60-89 from several villages in Doumen Town, Shaoxing City, Zhejiang Province, China. The inclusion criteria were: 1. Local residents with relatively stable work or residence; 2. No acute or chronic infectious diseases; 3. No progressive fatal diseases; 4. No serious mental illness; 5. No history of alcohol or drug abuse. The following situation was excluded: Those who had any physical or mental function problems and could not complete the survey. All participants in the study signed

written informed consent, and the elderly who would not sign pressed their fingerprints. The protocol and procedure of study were approved by the ethics committee of the First Affiliated Hospital of Zhejiang University (Reference Number: 20191276).

## Muscle mass measurement

Appendicular skeletal muscle mass (ASM) was measured by bio-electrical impedance analysis (BIA) (BCA-2A, Tsinghua Tongfang). Acknowledging the difficulty of measuring muscle mass in community settings, AWGS 2014 supported ASM measurement using bioelectrical impedance analysis (BIA) (*Chen et al., 2014*). AWGS 2019 continues this view. Before the test, subjects were asked to be fasting for 4 h and abstain from alcohol for 8 h. The subjects were wearing light clothes, standing barefoot on the instrument, legs slightly opened, hands held electrodes, arms were away from the trunk. Appendicular muscle mass (kg) divided by the square of height ($m^2$) was used to reflect the muscle mass ($ASM = kg/m^2$).

## Grip strength measurement

Jamar Hydraulic Hand Dynamometer (5030J1, Sammons Preston, Inc., Bolingbrook, Ill., USA) was used in this study. The standard positioning was sitting with 90 elbow flexion recommended by AWGS 2019. Took the maximum reading of at least 2 trials using either both hands or the dominant hand in a maximum-effort isometric contraction (*Chen et al., 2020b*).

## Demographic and health information

Every participant completed a survey questionnaire and trained researchers got answers from participants or their families through face-to-face interviews. The survey collected information on education level, personal income per year (thousands Chinese Yuan), smoking, drinking, chronic diseases, physical exercise and eating habits. Educational level was divided into three categories: illiterate, primary school, secondary school and above. Smoking status was divided into three categories: never smoking, current smoking and ever smoking. Drinking status was divided into three categories: never drinking, current drinking and ever drinking.

Chronic diseases were assessed by asking participants whether they had any of the following diseases or conditions, including: high blood pressure, diabetes, heart diseases, stroke or cerebrovascular disease, respiratory diseases (bronchitis, COPD and asthma), pulmonary tuberculosis, cataract, chronic kidney disease, cancer, gastrointestinal diseases, Parkinson's disease, falls, arthritis, dementia, metabolic disorder, cervical and lumbar diseases. According to the disease reported, number of chronic diseases was divided into four groups: no disease, one chronic disease, two chronic diseases, more than two chronic diseases, which was based on their distribution in the study sample (*Chen et al., 2020a*).

Physical exercise was assessed by asking whether taking regular exercises (seldom or often). Eating habits were assessed by asking the frequency of consumption of meat, fish, eggs, milk, beans, vegetables and fruits. For each aspect of eating habit, the reported frequency had four categories: do not eat, eat occasionally, often eat and eat every day.

## Statistical analysis

Continuous variables in accordance with normal distribution were expressed as means (standard deviation), and the comparison between the two groups was conducted by two independent sample $t$-test. The categorical variables were expressed by the number of cases and percentages, and the comparison between groups was performed by $\chi 2$ test. Prevalence ratio (PR) and 95% confidence interval (CI) were obtained by log binomial regression. In order to determine the related factors of the difference in the prevalence of sarcopenia between men and women, multivariate log binomial regression analysis was performed. $P < 0.05$ was determined as statistically significant. All the analysis were performed in STATA 14.

## RESULTS

We recruited 1,105 older adults, aged 60-89 years, including 557 men and 548 women. The mean age was 72.4 (0.31) years old in men, and 70.94 (0.30) years in women ($P < 0.001$, numbers in parentheses represent standard deviation). Figure 1 shows the screening process of sarcopenia in the study cohort.

### Participants characteristics

The following comparisons between groups were performed by $\chi 2$ test. The education levels and personal income per year of women were significantly lower than that of men ($P < 0.001$), women smoked less ($P < 0.001$) and drank less ($P < 0.001$) than men, women had more chronic diseases than men ($P = 0.001$). Compared to men, women ate less meat ($P < 0.001$), fish ($P < 0.001$), eggs ($P < 0.001$) and milk ($P = 0.004$). Women and men's eating habits of vegetables ($P = 0.154$) and fruits ($P = 0.343$) were similar; physical exercise habits were similar too ($P = 0.622$) (Table 1).

### Prevalence of sarcopenia, LGS and LMM

In the same way, sarcopenia, LGS and LMM were used as categorical variables and expressed in the number of cases and percentages, $\chi 2$ test was used for comparison between men and women. The overall prevalence of sarcopenia in this rural population was 17.29% (191/1105), with the prevalence in women (21.72%) was significantly higher than that in men (12.93%) ($P < 0.001$). The prevalence of low grip strength (LGS) and low muscle mass (LMM) were 46.50% (259/557) and 21.72% (121/557) in men, while 53.47% (293/548) and 27.01% (148/548) in women respectively ($P = 0.021$, $P = 0.041$) (Table 2).

### Associations between sex and sarcopenia, LGS and LMM

We then conducted log binomial regression analysis (Table 3) to evaluate the difference of prevalence of sarcopenia, LGS and LMM between men and women. Results showed that, compared to men, the proportion of women with sarcopenia was 1.68-fold greater (PR = 1.68, 95% CI [1.29–2.20], $P < 0.001$), LGS was 1.15-fold greater (PR = 1.15, 95% CI [1.02–1.29], $P = 0.021$) and LMM was 1.24-fold greater (PR = 1.24, 95% CI [1.01–1.53], $P = 0.041$) in women. After adjusting for age, education level and number of diseases, the associations for sarcopenia and LMM remained statistically significant. For example,

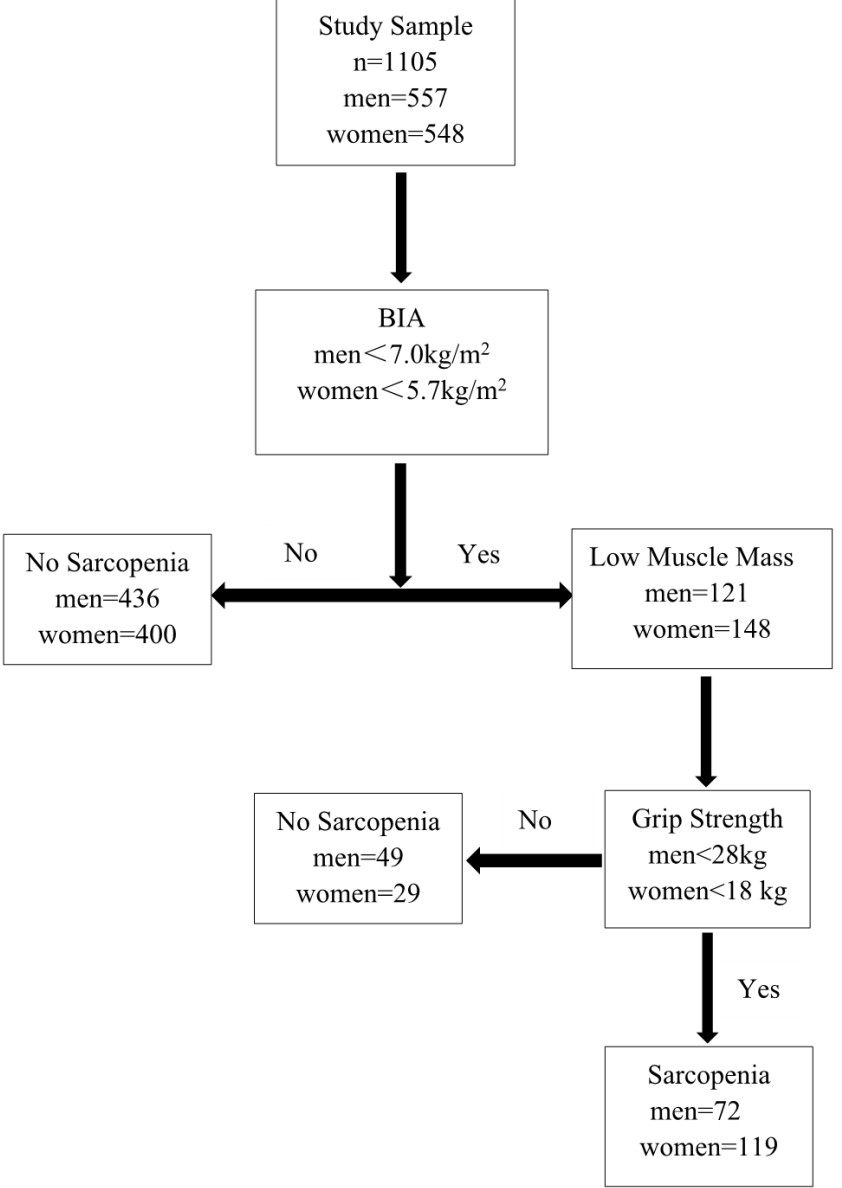

**Figure 1  Application of AWGS 2019 algorithm for the case finding of sarcopenia.**

the multivariable regression model (Multivariable Model 1) showed that women were more likely to have sarcopenia (PR = 1.61, 95% CI [1.19–2.18], $P = 0.002$) and LMM (PR = 1.31, 95% CI [1.03–1.66], $P = 0.028$). However, there was no difference in LGS between women and men (PR = 1.03, 95% CI [0.91–1.17], $P = 0.626$). On the basis of multivariable model 1, we further adjusted for income level, smoking status, drinking and eating habits (consumption of meat, egg, fish and milk) that differed between men and women (Multivariable Model 4). After adjusting above factors, the results showed that the prevalence of sarcopenia was 1.49-fold higher in women than in men (PR = 1.49,

**Table 1 Participants characteristics.** This table represents the baseline characteristic comparison of participants. The meaning of each data representation is reflected in the table.

| | Men<br>n = 557 | Women<br>n = 548 | P |
|---|---|---|---|
| Age (years), mean (SD) | 72.4(0.31) | 70.94(0.30) | <0.001 |
| Education level, n (%) | | | <0.001 |
|     Illiterate | 164(29.71) | 406(74.50) | |
|     Primary | 306(55.43) | 125(22.94) | |
|     Secondary and above | 82(14.86) | 14(2.57) | |
| Personal income per year (thousands CNY), median (IQR) | 25 (20–40) | 20 (10–25) | 0.001 |
| Number of diseases, n (%) | | | <0.001 |
|     No | 147(26.39) | 126(22.99) | |
|     1 | 259(46.50) | 212(38.69) | |
|     2 | 101(18.13) | 135(24.64) | |
|     More than 2 | 50(8.98) | 75(13.69) | |
| Smoking, n (%) | | | <0.001 |
|     Never | 269(48.47) | 540(99.08) | |
|     Current | 189(34.05) | 4(0.73) | |
|     Ever | 97(17.48) | 1(0.18) | |
| Drinking, n (%) | | | <0.001 |
|     Never | 203(36.51) | 442(80.95) | |
|     Current | 292(52.52) | 93(17.03) | |
|     Ever | 61(10.97) | 11(2.01) | |
| Physical exercise, n (%) | | | 0.622 |
|     Seldom | 437(78.88) | 424(77.66) | |
|     Often | 117(21.12) | 122(22.34) | |
| Meat, n (%) | | | <0.001 |
|     No | 49(8.81) | 202(36.93) | |
|     Occasionally | 193(34.71) | 205(37.48) | |
|     Often | 269(48.38) | 124(22.67) | |
|     Everyday | 45(8.09) | 16(2.93) | |
| Fish, n (%) | | | <0.001 |
|     No | 34(6.12) | 187(34.25) | |
|     Occasionally | 168(30.22) | 178(32.60) | |
|     Often | 337(60.61) | 177(32.42) | |
|     Everyday | 17(3.06) | 4(0.73) | |
| Eggs, n (%) | | | <0.001 |
|     No | 65(11.69) | 223(40.77) | |
|     Occasionally | 279(50.18) | 223(40.77) | |
|     Often | 189(33.99) | 88(16.09) | |
|     Everyday | 23(4.14) | 13(2.38) | |
| Milk, n (%) | | | 0.004 |
|     No | 278(50.00) | 332(60.81) | |
|     Occasionally | 154(27.70) | 114(20.88) | |

**Table 1** (*continued*)

| | Men n = 557 | Women n = 548 | P |
|---|---|---|---|
| Often | 81(14.57) | 69(12.64) | |
| Everyday | 43(7.73) | 31(5.68) | |
| Vegetables, n (%) | | | 0.154 |
| No | 4(0.72) | 3(0.55) | |
| Occasionally | 5(0.90) | 0(0.00) | |
| Often | 53(9.53) | 57(10.44) | |
| Everyday | 494(88.85) | 486(89.01) | |
| Fruits, n (%) | | | 0.343 |
| No | 56(10.09) | 50(9.17) | |
| Occasionally | 325(58.56) | 306(56.15) | |
| Often | 151(27.21) | 154(28.26) | |
| Everyday | 23(4.14) | 35(6.42) | |

**Notes.**
SD, standard deviation; CNY, Chinese Yuan; IQR, interquartile range.

**Table 2  Sex Differences in Sarcopenia, LGS and LMM.**

| | Men (n = 557) | Women (n = 548) | P |
|---|---|---|---|
| Sarcopenia, n (%) | | | <0.001 |
| Yes | 72(12.93) | 119(21.72) | |
| No | 485(87.07) | 429(78.28) | |
| LGS, n (%) | | | 0.021 |
| Yes | 259(46.50) | 293(53.47) | |
| No | 298(53.50) | 255(46.53) | |
| LMM, n (%) | | | 0.041 |
| Yes | 121(21.72) | 148(27.01) | |
| No | 436(78.28) | 400(72.99) | |

**Notes.**
LGS, low grip strength; LMM, low muscle mass.

95% CI [1.01–2.26], $P = 0.055$), and differences in LGS (PR = 0.96, 95% CI [0.81–1.14], $P = 0.641$) and LMM (PR = 1.14, 95% CI [0.83–1.55], $P = 0.422$) were not significant (Table 3).

## DISCUSSION

Using the AWGS 2019 sarcopenia diagnostic criteria, we found that the prevalence of sarcopenia was 1.68-fold greater in women (prevalence 21.72%) than in men (12.93%) among older adults from a rural area in eastern China. Adjusting for age, education level and number of diseases only slightly reduced the effect size for the prevalence of sarcopenia and low muscle mass but the association between sex and low grip strength became nonsignificant, suggesting that the difference in the prevalence of sarcopenia was mainly driven by the higher prevalence of low muscle mass in women rather than low grip strength. After further adjustment for income level, smoking, drinking and eating habits, the prevalence of sarcopenia in women was still 1.49-fold higher than men.

Yang et al. (2022), *PeerJ*, DOI 10.7717/peerj.13678

Peerj

**Table 3   Associations between sex and sarcopenia, LGS and LMM.** This table represents the prevalence of sarcopenia, low muscle mass and low grip strength in men and women under different five models before and after adjustment for confounders. The significance represented by each data is represented in the table.

| | Univariable model | *P* | Multivariable model 1 | *P* | Multivariable model 2 | *P* | Multivariable model 3 | *P* | Multivariable model 4 | *P* |
|---|---|---|---|---|---|---|---|---|---|---|
| | PR (95% CI) | | PR (95% CI) | | PR (95% CI) | | PR (95% CI) | | PR (95% CI) | |
| **Associations between sex and sarcopenia** | | | | | | | | | | |
| Men | 1 (reference) | | 1 (reference) | | 1 (reference) | | 1 (reference) | | 1 (reference) | |
| Women | 1.68 (1.29–2.20) | <0.001 | 1.61(1.19–2.18) | 0.002 | 1.42 (1.04–1.93) | 0.026 | 1.58 (1.07–2.32) | 0.021 | 1.49 (1.01–2.26) | 0.055 |
| **Associations between sex and LGS** | | | | | | | | | | |
| Men | 1 (reference) | | 1 (reference) | | 1 (reference) | | 1 (reference) | | 1 (reference) | |
| Women | 1.15 (1.02–1.29) | 0.021 | 1.03(0.91–1.17) | 0.626 | 0.95 (0.83–1.08) | 0.398 | 0.99 (0.95–1.16) | 0.921 | 0.96 (0.81–1.14) | 0.641 |
| **Associations between sex and LMM** | | | | | | | | | | |
| Men | 1 (reference) | | 1 (reference) | | 1 (reference) | | 1 (reference) | | 1 (reference) | |
| Women | 1.24 (1.01–1.53) | 0.041 | 1.31(1.03–1.66) | 0.028 | 1.19 (0.93–1.52) | 0.172 | 1.09 (0.81–1.47) | 0.586 | 1.14 (0.83–1.55) | 0.422 |

**Notes.**

PR, prevalence ratio; CI, confidence interval; LGS, low grip strength; LMM, low muscle mass.

Multivariable Model 1: adjusted for age, education level and number of diseases. Multivariable Model 2: adjusted for age, education level and number of diseases and income level. Multivariable Model 3: adjusted for age, education level, number of diseases, income level, smoking status, drinking alcohol, and physical exercise. Multivariable Model 4: adjusted for age, education level, number of diseases, income level, smoking status, drinking alcohol and eating habits of fish, milk, egg and meat.

Our study found that elderly women were more likely to have sarcopenia than men in a rural area in eastern China. However, the review of epidemiology studies conducted in Asian countries reported that the prevalence of sarcopenia was more predominant in men than in women (5.1%–21.0% in men vs 4.1%–16.3% in women) (*Chen et al., 2020b*). The different source of population might explain the different prevalence across studies. While there were limited epidemiological studies assessing prevalence of sarcopenia that specifically restricted to older adults from rural areas in China, it was reported that older adults from rural area in China were much more likely to have worse health outcomes such as malnutrition and frailty (*Dai, Zhou & Mei, 2013*). Compared to men, older women from rural areas were more disadvantaged as they generally had lower education level and lower income than men (*Dai, Zhou & Mei, 2013*). It has been reported that the prevalence of disability was higher in women than in men in older adults from rural China (*Stewart Williams, Norström & Ng, 2017*). The prevalence of sarcopenia increases gradually with age (*Martinez et al., 2015*; *Yu et al., 2014*), and is associated with increased risk of falls, lower quality of life and many chronic diseases such as diabetes, cardiovascular disease, heart failure, renal insufficiency, cancer, cognitive impairment, Parkinson's syndrome and depression, and even the prognosis of chronic diseases (*Chen et al., 2016*; *Hsu et al., 2014*; *Srikanthan & Karlamangla, 2011*). To maintain better health status and eliminate gender inequality in health, it is important to provide early screening and effective interventions for sarcopenia in older female adults in the rural community of China.

Adjusting for sociodemographic characteristics, number of diseases and lifestyle behavior factors in the multivariable model reduced the association, suggesting that the difference in prevalence of sarcopenia was partially explained by these factors. But the remained 1.49-fold greater prevalence in women than men suggested that there were other factors other than the above factors strongly driving this difference. While smoking was shown to be a risk factor for sarcopenia (*Rom et al., 2012*; *Steffl et al., 2015*), and the proportion of people smoking was much higher in men than in women, we still observed significant higher prevalence of sarcopenia in women after adjusting for smoking.

In our study, we only used simple questions to measure diet and physical exercise so that we were unable to quantify nutrition intake from diet and physical exercise level from different domains. Thus, our results should be interpreted with caution that the mediating effects of diet and physical exercise on the association between sex and sarcopenia could not be excluded. Adequate intake and absorption of energy and protein are important for older adults to maintain muscle health (*Komar, Schwingshackl & Hoffmann, 2015*; *Liao et al., 2017*; *Lord et al., 2007*). Studies have shown nutritional deficiency increased the risk of sarcopenia (*Molnár et al., 2016*; *Naseeb & Volpe, 2017*).

Protein, vitamin D and calcium supplementation can improve skeletal and muscle health in postmenopausal women (*Rizzoli et al., 2014*). In our study cohort, we found that women reported less frequency intake of eggs, milk, fish and meats, which are the major source of protein intake from foods. Future studies using food frequency questionnaire or dietary recall to quantify the specific nutrient intake are needed to assess whether and to what extent poor diet quality explained the higher prevalence of sarcopenia in women.

In addition, elderly people who lack physical exercise are more likely to experience skeletal muscle mass loss and muscle strength reduction, thus leading to increased risk of developing sarcopenia (*Burton & Sumukadas, 2010*; *Naseeb & Volpe, 2017*). Aerobic, balance, strength, flexibility, and ball sports were included in the questionnaire regarding physical activity forms. We found that the physical activities of the rural population were basically walking, domestic and farm works, and there were almost no recreational sports such as ball games, yoga or gym, which may be related to the fact that the research objects were rural people. Therefore, we divided the variable physical activity simply into two categories of seldom and often. A study showed that the rural population in Western China was more prone to sarcopenia than the urban population, which was independent of physical activity (*Gao et al., 2015*). Another study from Brazil also suggested that the difference in the prevalence of sarcopenia between rural and urban women was not related to physical activity (*Mazocco et al., 2019*). A large-scale longitudinal study conducted in nine different provinces and three mega cities (Beijing, Shanghai and Chongqing) in China from 1989 to 2011 included people from rural, urban and suburban areas with significant differences in geography, economic development, public resources and health indicators. The results showed that throughout the 20-year period, for both adult men and adult women in China, occupational work and domestic labor were the largest contributors to physical activity, while active leisure and travel activities were the small ones (*Ng et al., 2014*). The statistical results of our study regarding physical activity were also consistent with the conclusion. In our study cohort, around 80% reported no regular exercise in both men and women. Since the majority older adults from rural areas in China were traditional Chinese farmers, and we did not measure physical exercise from walking, domestic and farm works, it was possible that men involved more in physical-demanding farm works or other works than women, which may benefit them with better muscle health.

There were some other factors associated with the development of sarcopenia that were not assessed in the current study. Age-related changes in the endocrine system and hormone levels play an important role in the pathogenesis of sarcopenia. For example, testosterone could increase muscle mass and enhance muscle function, and reduction in testosterone level is involved in the development of sarcopenia (*Herbst & Bhasin, 2004*). It has been observed that androgen supplements play an important role in promoting muscle strength and increasing muscle mass. Recently, a study on 94 elderly people with normal thyroid function found that higher concentration of free triiodothyronine (FT3) in the normal range was positively correlated with the muscle mass and muscle function of the elderly (*Sheng et al., 2019*). Other factors related to sarcopenia include motor neuron degeneration (*Drey et al., 2014*), genetic factors (*Tan et al., 2012*), inflammatory factor (*Frost & Lang, 2007*; *Patel et al., 2014*) and insulin resistance (*Kwon et al., 2017*; *Moon, 2014*). Gender difference has been found in the process of aging (*Marais et al., 2018*; *Thomas, Gurvich & Kulkarni, 2019*). Whether the above factors explained the observed difference in the prevalence of sarcopenia between men and women needs further researches.

The strengths of our study include that we used the latest diagnostic criteria for sarcopenia from AWGS 2019 that combined low grip strength and low muscle mass. To date, there were few studies evaluating sarcopenia using the updated criteria. Secondly, we specifically

focused on older adults from rural areas in Zhejiang Province with large sample size. There were several limitations need to be acknowledged of the current study. First of all, we used convenient sample that were recruited in Doumen Town, which might not be representative of the general rural older residence in Zhejiang Province, China. In addition, we did not use detailed validated questionnaire to quantify nutrition intake from diet and physical activity, which limited us to comprehensively assess their effects on the observed associations. Lastly, for chronic diseases, we used the count of self-reported diseases, which did not capture the severity of diseases.

# CONCLUSIONS

In conclusion, according to the AWGS 2019 diagnostic criteria, the prevalence of sarcopenia was higher in elderly women than that in men in a rural area of eastern China. The finding suggests that women in rural areas in China seem more vulnerable for sarcopenia, thus early screening and prevention need to be provided for them to address such gender disparity in health. Adjusting for sociodemographic, chronic diseases and lifestyle behaviors reduced the associations but the prevalence of sarcopenia in women remained 1.49-fold greater than in men, suggesting that there were other factors strongly driving the difference. To provide more helpful information for health policy makers and to develop more tailored intervention programs, further studies are needed to explore the possible reasons for such difference.

# ACKNOWLEDGEMENTS

We appreciated the community staff of Doumen Town for their testimony and other contributions to the smooth progress of the study.

## Funding

This study was supported by the National Key Research and Development program of China (No. 2018YFC2000301, Funder: Qin Zhang), the Key R&D Program of Zhejiang (No. 2022C03161, Funder: Qin Zhang) and the Medical Health Science and Technology Project of Zhejiang Provincial Health Commission (No. 202038269, Funder: Danfeng Deng; No. 2021430262, Funder: Yuming Wang). There was no additional external funding received for this study. The funders had no role in study design, data collection and analysis, decision to publish, or preparation of the manuscript.

## Grant Disclosures

The following grant information was disclosed by the authors:
National Key Research and Development program of China: 2018YFC2000301.
Key R&D Program of Zhejiang: 2022C03161.
Medical Health Science and Technology Project of Zhejiang Provincial Health Commission: 202038269, 2021430262.

## Competing Interests

The authors declare there are no competing interests.

## Author Contributions

- Yichen Yang performed the experiments, prepared figures and/or tables, authored or reviewed drafts of the article, and approved the final draft.
- Qin Zhang conceived and designed the experiments, authored or reviewed drafts of the article, and approved the final draft.
- Caihong He performed the experiments, authored or reviewed drafts of the article, and approved the final draft.
- Jing Chen analyzed the data, authored or reviewed drafts of the article, and approved the final draft.
- Danfeng Deng performed the experiments, prepared figures and/or tables, and approved the final draft.
- Wenwen Lu performed the experiments, prepared figures and/or tables, and approved the final draft.
- Yuming Wang performed the experiments, prepared figures and/or tables, and approved the final draft.

## Human Ethics

The following information was supplied relating to ethical approvals (i.e., approving body and any reference numbers):

First Affiliated Hospital of Zhejiang University granted Ethical approval to carry out the study within its facilities (Reference Number: 20191276).

## Data Availability

The raw data are available in the Supplementary File.

## Supplemental Information

Supplemental information for this article can be found online at http://dx.doi.org/10.7717/peerj.13678#supplemental-information.

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
