# Peer review of "Prevalence of sarcopenia was higher in women than in men: a cross-sectional study from a rural area in eastern China"

_PeerJ, doi:10.7717/peerj.13678_

## Round 0.1 · original submission · Minor Revisions

The three reviewers and I are impressed with many aspects of your study and manuscript. However, the reviewers have identified a number of areas in which the manuscript can be improved. Please look to address these concerns prior to resubmitting the manuscript.

Reviewer 1 ·

Basic reporting

Please see general comments

Experimental design

Please see general comments

Validity of the findings

Please see general comments

Additional comments

This manuscript is a straight forward descriptive study of the prevalence of sarcopenia in a rural area in China, documenting that the prevalence in women is higher than in men, even after adjusting for co-variates. This is a finding that is contrary to the majority of studies.

The manuscript could be improved in the following ways:
1) Literature review: please include references for gender differences in urban settings, and among non-Chinese populations. In particular please include prevalence studies in urban settings in China: for example there have been many studies on this topic from the West China Medical University, directed by Prof Dong Birong.
2) Please show the response rate of all the eligible population, and reason(s) for non-response, so that any potential bias may be detected.
3) Please show the prevalence of sarcopenia for men and women, before and after adjustment for confounding factors.
4) Please show the factors associated with sarcopenia (using variables shown in Table 1) using logistic regression, separately for men and women.

·

Basic reporting

The writing of this article is clear in English. It provides adequate background info with relevant and sufficient references. The data structure including figures and tables is appropriate.

Experimental design

This is a national multi-center study in China. The data reported in this paper are cross-sectional with relevant survey info and objective measures. The research question is clearly defined and scientific approach is vigorous and described in detail with no ethical concern.

Validity of the findings

Sex and gender difference in sarcopenia is important with potentially high impact. This article reports underlying data that are robust to address this research question. The conclusion is well stated and supportive of the original hypothesis.

Additional comments

One suggestion for the "Conclusions" section (in both Abstract and at the last section of the text) is to change "eliminate" to "address" in the statement " ...to eliminate such gender disparity in health." as such disparity would likely be completed eliminated.

Reviewer 3 ·

Basic reporting

1) Thank you for providing the raw data, however the data should be in excel format instead of stata format for easy viewing

2) The english can be improved by checking through subject verb agreements e.g. line 31 should be "women in rural areas in China seem more vulnerable…” and line 37 “decline in quality of life".

Experimental design

No comment

Validity of the findings

No comment

Additional comments

1) In the abstract section under results, “the prevalence of sarcopenia was 151% times higher…” this sentence be rephrased as proportion of people with sarcopenia is 1.5-fold greater... (95%CI 1.01-2.26, P = xxx), with the actual p value stated. Please do the same for the results section lines 139-142.

2) Lines 40-41 seems to be linked to 36-37. Suggest to rephrase into 1 sentence.

3) Lines 98-100 - it is unclear why the authors divided multimorbidity into 4 groups rather than presence/absence of multimorbidity or number of of chronic diseases as a continuous variable.

4) Lines 102 - While the types and time per week for physical activity was determined, this was not reflected in Table 1 where the physical activity variable was binary.

5) Lines 117-118 - “Two-sample t-test…mean age were 72.4(0.31) years…” Please check and rephrase this sentence as t-test should not be needed for describing the mean age, provided there is a difference between groups, which in that case, the p values should be provided. Please also describe what the numbers in the brackets mean e.g. SD?

6) In Table 3, please state what each model represents for easy reference at the footnote

7) The definition of physical exercise is vague in the study - does this include non-recreational physical activity such as transportation, occupational, housework? Non-recreational physical activity has been shown to be higher in less developed areas, compared to developed areas where people engage largely in recreational physical activity (due to transition towards sedentary occupations and motorised vehicles etc.) This should perhaps be included in the discussion.

8) Was socio-economic status measured in this study? If so, it should be included in the descriptive statistics, as the authors mentioned that women are disadvantaged in terms of income levels - which might also affect a myriad of lifestyle factors and hence sarcopenia.

---

## Round 0.2 · Minor Revisions

In addition to responding to the minor amendments requested by reviewer three, please do the following amendments, with these line numbers based on the track changes word document:

Line 47-50: please write out in full all of these acronyms for the different sarcopenia diagnostic approaches and ensure all other acronyms (such as those described in the methods and results) and used in the rest of the manuscript are also clearly defined in the first instance they are used after the abstract.

Line 236: please remove mention of Professor Birong Dong’s here and anywhere else in the manuscript. Just provide the relevant references.

Reviewer 3 ·

Basic reporting

No comment

Experimental design

No comment

Validity of the findings

No comment

Additional comments

1. Since income levels (socioeconomic status) differed between men and women, this should be accounted for in the subsequent analyses.

2. Please provide a reference for the adjustment of co-morbidities in a non-linear manner as the authors have done in this paper.

3. In the introduction and discussion sections, it is not necessary to mention the Professor's name as these would be covered in the cited references.

4. The literature gaps need to be identified and justified clearly in the introduction. In the discussion, the authors mentioned Gao et al 2015 has also investigated sarcopenia prevalence in rural western China. How does this present study differ from earlier studies? What new knowledge is added?

---

## Round 0.3 · Minor Revisions

I think the authors for attending to most of the constructive criticisms raised earlier. Some small things which still require amendments include the following, with all page numbers reflecting the most recent version of the track changes manuscript.

Line 46: remove the empty space in the following ( IWGS);
Line 48: provide a space between Sarcopenia(AWGS);
Line 128: there is an empty comment by Jing Chen, can this be removed?
Reviewer threes comments, Point 2: it is unclear if you have provided a reference to the adjustment of comorbidities in your analyses. If you have done so, can you please make that more explicit that such an amendment has been made.

---

## Round 0.4 · accepted · Accept

I think the authors for their diligence in attending to the initial concerns of the reviewers and I. I would like to recommend this paper be accepted for publication in PeerJ.